# Skin *Cutibacterium acnes* Mediates Fermentation to Suppress the Calcium Phosphate-Induced Itching: A Butyric Acid Derivative with Potential for Uremic Pruritus

**DOI:** 10.3390/jcm9020312

**Published:** 2020-01-22

**Authors:** Sunita Keshari, Yanhan Wang, Deron Raymond Herr, Sung-Min Wang, Wu-Chang Yang, Tsung-Hsien Chuang, Chien-Lung Chen, Chun-Ming Huang

**Affiliations:** 1Department of Life Sciences, National Central University, Taoyuan 32001, Taiwan; sunitakeshari827@gmail.com; 2Department of Dermatology, University of California, San Diego, CA 92093, USA; yaw015@ucsd.edu; 3Department of Pharmacology, National University of Singapore, Singapore 117600, Singapore; phcdrh@nus.edu.sg; 4Department of Biomedical Sciences and Engineering, National Central University, Taoyuan 32001, Taiwan; sss810621@yahoo.com.tw; 5Division of Nephrology, Landseed International Hospital, Taoyuan 32001, Taiwan; wuchang.yang2@gmail.com; 6Immunology Research Center, National Health Research Institutes, Zhunan, Miaoli County 350, Taiwan; thchuang@nhri.edu.tw

**Keywords:** CaP, CKD, *C. acnes*, fermentation, BA-NH-NH-BA

## Abstract

Pruritus and inflammation associated with accumulation of calcium phosphate (CaP) under the skin are common problems among dialysis patients with chronic kidney disease (CKD). The role of skin commensal microbiota in the CaP-induced uremic pruritus remains uncharacterized. Skin *Cutibacterium acnes* (*C. acnes*) can solubilize CaP by the production of short-chain fatty acids (SCFAs), such as butyric acid, through glucose fermentation. Like butyric acid, the N-[2-(2-Butyrylamino-ethoxy)-ethyl]-butyramide (BA-NH-NH-BA), a butyric acid derivative, remarkably induced acetylation of histone H3 lysine 9 (AcH3K9) in keratinocytes. Topical application of fermenting *C. acnes*, butyric acid or BA-NH-NH-BA onto mouse skin effectively ameliorated CaP-induced skin itching, interleukin (IL)-6 up-regulation in keratinocytes, and extracellular signal-regulated kinase (ERK) 1/2 activation in dorsal root ganglia (DRG). Activation of ERK 1/2 by CaP was markedly reduced in IL-6 knockout mice. Genus *Cutibacterium* was detected in relatively low abundance in itchy skin of patients with CKD. Our results identify a role for the skin fermenting *C. acnes* in ameliorating CaP-induced activation of IL-6/p-ERK signaling and resulting skin inflammation. Furthermore, we provide evidence for the potential therapeutic efficacy of BA-NH-NH-BA as a postbiotic for the treatment of uremic pruritus.

## 1. Introduction

Pruritus is a condition characterized by severe itching and has a profound impact on the quality of life of affected individuals [1]. It can be acute or chronic, resulting from xerosis or eczema, cirrhosis, hematologic disorders, infection, or drug reactions. Pruritus is one of the most distressing symptoms in chronic kidney disease (CKD) [2] also termed, “CKD associated pruritus” (CKD-P) [3]. Uremic pruritus is extremely common in patients undergoing hemodialysis [4,5]. For example, a recent study of 18,801 hemodialysis patients demonstrated that pruritus affected approximately 42% of all patients, and 15–49% of patients with chronic renal failure. Notably, 50–90% of the dialysis population experienced significant itch [6,7,8]. 

CKD represents a major public health burden and its incidence is likely to increase as the population ages [9]. Currently, CKD is the ninth leading cause of death in the United States (US) affecting up to 14% of the adult population and 25% of those are ≥ 60 years old [10]. CKD mostly involves imbalanced regulation of minerals and hormones, such as calcitriol, calcium, phosphate, and parathyroid hormone (PTH), and this imbalance is likely to underlie the development of uremic pruritus [11]. The Dialysis Outcomes and Practice Patterns Study (DOPPS) showed strong independent relationships between higher serum calcium (> 0.102 mg/mL), phosphorus (> 0.055 mg/mL), and calcium phosphorus (CaP) product (> 0.8 mg^2^/mL^2^) in uremic pruritus incidence [12,13]. CaP product above 70 mg^2^/mL^2^ is found to be the major cause of metastatic calcification in soft tissue in patients with CKD [14,15]. Moreover, it has been reported that the levels of calcium, as well as CaP in the epidermal layer of skin in hemodialysis patients with pruritus, are significantly higher than those in hemodialysis patients without pruritus or healthy subjects [2,16,17].

Inflammatory signaling pathways represent important mechanistic components of CKD. For example, G protein-coupled receptor (GPCR)-mediated central sensitization of itch signaling in the spinal cord and brain levels contribute significantly to abnormal modulation of chronic itch against pruritogens in different diseases [18,19]. This is likely to involve extracellular signal-regulated kinases (ERK) signaling activation, which plays an active role in elevating spinal neuron excitability in response to pruritogens like dinitrofluorobenzene (DNFB) [20,21]. Peripheral sensitization is likely to involve inflammatory cytokines. Increased interleukin (IL)-6 in serum and skin is associated with mortality in patients with CKD, suggesting that inflammatory processes could be ameliorated by targeting IL-6 [22,23]. Furthermore, overexpression of T helper (Th)2 cell-induced IL-31 in mice leads to chronic itch and inflammation in atopic dermatitis (AD) and psoriasis where therapeutic benefits have been observed by blocking with anti-IL-31 antibodies [24,25,26]. Further, IL-31 activated phosphorylated ERK (p-ERK) 1/2 in dorsal root ganglion (DRG) of a mouse model of atopic-like dermatitis, inhibition of which blocked IL-31 signaling in vitro and reduced IL-31-induced scratching in vivo suggesting that ERK 1/2 activation in the DRG is an integral phenomenon in neural transmission mediation of the itch sensation [27].

Histone deacetylases (HDAC) inhibition also resulted in a dose-dependent reduction of IL-6-induced phosphorylation of signal transducer and activator of transcription 3 (STAT3) in naïve cluster of differentiation (CD4^+^) T cells [28]. Moreover, inhibition of HDAC reduces the level of urinary markers of lupus and IL-6 and improves kidney histopathology in lupus nephritis (LN) [29]. These studies reveal a significant role of the HDAC regulation in inflammation in kidney disease. However, the molecular mechanisms of HDAC concerning the regulation of IL-6 in chronic pruritus remain elusive.

Human skin is colonized by millions of microorganisms after birth. *Cutibacterium acnes* (*C. acnes*) (previously known as *Propionibacterium acnes*) accounts for approximately half of the total skin microbiome with an estimated density of 10^2^ to 10^6^ per cm^2^ [30,31]. Frequent colonization by *Staphylococcus aureus* (*S. aureus*), with a lower abundance relative to that of *C. acnes*, is implicated in the pathogenesis and severity of both AD and psoriasis [32,33,34]. Studies have shown that a decrease in immunoregulatory bacteria such as *C. acnes* may lead to higher colonization of *S. aureus*, which could exacerbate cutaneous inflammation in these diseases [35,36,37]. Short-chain fatty acids (SCFAs), including butyric acid, from carbohydrate fermentation of *C. acnes* or *Staphylococcus epidermidis* (*S. epidermidis*) can help ease inflammation in infected skin [38,39]. More directly, butyric acid has been specifically identified as an active itch-inhibiting compound in AD and psoriasis [40,41]. 

We previously demonstrated that chronic CaP deposition in skin led to pruritus with upregulated IL-6, resulting in ERK 1/2 activation in the DRG of the spinal dorsal horn of Institute for Cancer Research (ICR) mice [42]. In the current study, we explored whether CaP-induced pruritic inflammation and ERK 1/2 activation in the DRG can be regulated by fermentation of skin commensal bacteria. Here we show that *C. acnes* fermentation can influence CaP-induced IL-6 in skin and ERK 1/2 activation in DRG through epigenetic mechanisms and that a synthesized butyric acid derivative can efficiently attenuate CaP-induced inflammation and pruritus.

## 2. Materials and Methods

### 2.1. Ethics Statement

This study was conducted in accordance with the protocols (NCU-106-016, 19 December 2017) of the Institutional Animal Care and Use Committee (IACUC) of National Central University (NCU). ICR, IL-6 knockout (KO), and wild-type (WT) C57BL/6 mice (8–9-week-old females; National Laboratory Animal Center, Taipei, Taiwan) were sacrificed by asphyxiation with CO_2_ [42]. The Institutional Review Board (IRB) (No. 19-013-B1, 22 May 2019) at Landseed International Hospital approved the consent procedure for skin swab sampling. Skin swabs were collected from healthy subjects (*n* = 10) and non-itchy (*n* = 10) and itchy (*n* = 10) skin of chronic hemodialysis patients with CKD beyond 50 years old. Written consent was obtained from all participants prior to inclusion in the study. 

### 2.2. Bacterial Culture and Fermentation

Chemicals were purchased from Thermo Fisher Scientific (Fair Lawn, NJ, USA) or companies as indicated. Human skin bacteria were collected by sterile swabs (Becton, Dickinson and Company, Franklin Lakes, NJ, USA) along the surface of a healthy subject and spread onto a CaP (5 g/L)-rich Pikovskaya’s agar plate supplemented with 2% glucose for 3 days. Bacteria identified as *C. acnes* or *B. shackletonii* were cultured to an absorbance at 600 nm [optical density (OD)_600_] of 1.0. Bacteria were harvested by centrifugation at 5000 × g for 10 min, washed with and suspended in phosphate-buffered saline (PBS). For fermentation, bacteria (10^5^ CFU/mL) were incubated in rich media in the absence and presence of 2% glucose under anaerobic conditions using the Gas-Pak at 37 °C. Rich media plus 2% glucose without bacteria was used as controls. 0.001% (w/v) phenol red (Sigma, Burlington, MA, USA) in rich media was used as an indicator, which changes from red-orange to yellow when fermentation occurs. 

### 2.3. GC-MS Analysis

*C. acnes* (10^5^ CFU/mL) was incubated in rich media with 2% glucose for 3 days. The fermentation media was centrifuged at 5000 × g to remove *C. acnes*. The fermentation media was separately subjected to liquid-liquid extraction with ethyl acetate (Residue Analysis OmniSolv, Millipore, Billerica, MA, USA) after adding an internal standard (0.1 mg/mL ^2^H7-butyric acid, C/D/N Isotopes, Pointe-Claire, Quebec, Canada), followed by acidification with 0.5% ortho-phosphoric acid and saturation with sodium chloride. Gas chromatography-mass spectrometry (GC-MS) analysis was performed using an Agilent 5890 Series II GC coupled with a 5971 MS detector (Agilent Technologies, Inc., Palo Alto, CA, USA). A 70-eV electron beam was used for ionization. SCFAs in the fermentation media were quantified by a calibration curve made from six non-zero levels using the Free Fatty Acids Test Standard (Restek Corporation, Bellefonte, PA, USA).

### 2.4. CaP Degradation

CaP (5 mg/mL) was mixed with media from the glucose fermentation of *C. acnes* or media from the culture of *C. acnes* or glucose alone for 24 h. All mixtures were filtered using a mixed cellulose esters membrane (5 μm pore size; Sigma, Burlington, MA, USA) and kept in rotation overnight. Then, 1000 μL/cuvette of each sample was used to determine the particle size by dynamic light scattering (DLS) (Malvern Instruments Ltd., Malvern, UK). A total of 20, 50, 100, and 200 mM of acetic, propionic, or butyric acids were dropped onto a Pikovskaya’s agar plate. The clear zone of CaP on the agar plate was measured.

### 2.5. RT-PCR

CCD 1102 KERTr cells (ATCC CRL-2310) were grown in Defined Keratinocyte-Serum Free Media supplemented with bovine pituitary extract (BPE), epidermal growth factor (EGF), and antibiotics. Different groups of KERTr cells were treated with water, acetic acid, propionic acid, butyric acid, or BA-NH-NH-BA (4 mM) in the presence or absence of CaP for 24 h. Total cellular RNA was extracted, reverse transcribed to cDNA using iScript cDNA synthesis kit (Bio-Rad, Hercules, CA, USA) and amplified by RT-PCR on the CFX96 real-time system (Bio-Rad, Hercules, CA, USA). The comparative cycle threshold (∆∆CT) was used to determine the quantification of gene expression. The gene level of glyceraldehyde 3-phosphate dehydrogenase (GAPDH) was used for the normalization of IL-6 expression. The 16S gene expression of *C. acnes* was measured relative to *Staphylococcus* species. Primers for IL-6 and GAPDH were 5′-CCCAGGAGAAGATTCCAAAGAT-3′ (forward); 5′-CTGGCTTGTTCCTCACTACTC-3′ (reverse) and 5′-GACTTCAACAGCAACTCCCAC-3′ (forward); 5′-TCCACCACCCTGTTGCTGTA-3′ (reverse), respectively. Primers for detection of 16S rRNA gene of *C. acnes* and *Staphylococcus* species were: 5′-ATACGTAGGGTGCGAGCGTTGTCC-3′ (forward); 5′-TGGTGTTCCTCCTGATATCTGCGC-3′ (reverse) and 5′-TTTGGGCTACACACGTGCTACAATGGACAA-3′ (forward); 5′-AACAACTTTATGGGATTTGCWTGA-3′ (reverse), respectively.

### 2.6. Administration of CaP into Cells and Mice

CaP (5 mg/mL) was added to KERTr cells for 10 min before the addition of 4 mM of acetic acid, propionic acid, butyric acid, or BA-NH-NH-BA, which was synthesized using a published protocol [41]. Human KERTr or HaCaT cells were used because they produced a plethora of cytokines including interleukin (IL)-1, -6, -7, -8, -10, -12, -15, -18, and -20, and tumor necrosis factor (TNF)-α during skin inflammation. These cells have been widely used for studies of drug responses and skin innate immunity [43,44]. BA-NH-NH-BA was topically applied on the dorsal skin of ICR mice 10 min before CaP injection. All SCFAs, BA-NH-NH-BA, and CaP were dissolved in water. CaP (5 mg/mL) was injected intradermally into the dorsal skin of mice to generate CKD-like pruritic conditions. *C. acnes* (10^5^ CFU/mL) and 2% glucose in PBS were topically applied to the dorsal skin of mice for 3 days, followed by CaP injection on the third day. Scratching behavior was measured for 2 h. The scratching response to CaP was evaluated by counting the numbers of spontaneous scratches occurring to the injection site by a hind paw during the test period. Protocols for detection of secretion of IL-6 by ELISA, the expression of p-ERK 1/2, GAPDH, or β-actin by western blotting, and AcH3K9 by immunostaining were described in detail in Appendix A.

### 2.7. Flow Cytometry

The dorsal skin of C57BL/6 mice was topically applied with 100 μL of PBS or 4 mM acetic acid, butyric acid or BA-NH-NH-BA in PBS for 2 h. Single-cell suspensions were prepared from skin biopsy by mincing tissue with scissors, followed by enzymatic digestion for 1 h with 2.5 mg/mL collagenase Type II and Type IV (Life Technologies, Grand Island, Carlsbad, CA, USA), and 0.5 mg/mL DNase (Roche, Indianapolis, IN, USA) in PBS containing 1% bovine serum albumin (BSA) (Abcam, Cambridge, UK) at 37 °C. Digests were quenched by adding Dulbecco’s modified Eagle medium (DMEM) containing 10% fetal bovine serum (FBS) and subsequently filtered through a 70 μm nylon filter (BD Biosciences, San Jose, CA, USA). Cells were washed with DMEM containing 10% FBS, followed by washing with PBS/BSA. Cells were fixed with 0.01% formaldehyde (Polyscience, Warrington, PA, USA) and then were permeabilized with 0.3% Triton (Sigma, Burlington, MA, USA) before incubation with Alexa Fluor^®^ 647 anti-acetylated lysine (Biolegend, San Diego, CA, USA) and Alexa Fluor^®^ 488 anti-K10 antibody [EP1607IHCY] (Abcam, Cambridge, UK) at a dilution of 0.1 μg/ml in PBS/BSA for 1 h at 4 °C. Cells were analyzed with the Guava EasyCyte 8HT two-laser, six-color microcapillary-based benchtop flow cytometer (Millipore, Billerica, MA, USA) for assessing AcH3K9 and K10 expression. The flow cytometry profile represents the staining of AcH3K9 on the x-axis and K10 on the y-axis.

### 2.8. NGS Analysis

Total DNA was extracted from skin swab samples of non-itchy and itchy skin of patients undergoing chronic dialysis using a QIAamp DNA Stool Mini Kit (Qiagen, Hilden, Germany) [45]. NGS was performed for ten samples from each experimental group (*n* = 20). The eubacterial primer kit (Qiagen, Hilden, Germany) was used under the following conditions: 94 °C for 3 min, followed by 28 cycles of 94 °C for 30 s; 53 °C for 40 s and 72 °C for 1 min; after which a final elongation step at 72 °C for 5 min was performed. Following PCR, all amplicon products from different samples were mixed in equal concentrations and purified using Agencourt Ampure beads (Agencourt Bioscience Corporation, United Beverly, MA, USA). Samples were sequenced utilizing Roche 454 FLX titanium instruments (Roche, Indianapolis, IN, USA). The 16S rRNA gene V4 variable region PCR primers 515/806 [46] were used in a single-step 30 cycle PCR using the HotStarTaq Plus Master Mix Kit (Qiagen) under the following conditions: 94 °C for 3 min, followed by 28 cycles of 94 °C for 30 s, 53 °C for 40 s and 72 °C for 1 min, after which a final elongation step at 72 °C for 5 min was performed. Sequencing was performed at MR DNA (www.mrdnalab.com, Shallowater, Lubbock, TX, USA) on an Ion Torrent Personal Genome Machine (PGM). Sequence data were processed using a proprietary analysis pipeline (MR DNA). All sequences were depleted of barcodes and primers, then sequences < 150 bp, with ambiguous base calls or homopolymer run exceeding 6 bp were removed. Sequences were denoised, 16S rRNA operational taxonomic units (OTUs) were generated, and chimeras removed. OTUs were defined by clustering at 3% divergence (97% similarity). Final OTUs were taxonomically classified using BLASTn against a curated database derived from GreenGenes, RDPII, and NCBI (www.ncbi.nlm.nih.gov, http://rdp.cme.msu.edu) [46,47].

### 2.9. Statistical Analysis

Data analysis was performed using an unpaired *t-*test or one-way ANOVA with GraphPad Prism^®^ software. The *p*-values of < 0.05 (*), < 0.01 (**), and < 0.001 (***) were considered significant. The mean ± standard error (SE) was calculated for at least three independent experiments.

## 3. Results

### 3.1. C. acnes Fermentation and CaP Solubilization by Fermentation Metabolites

A CaP (5 g/L)-rich Pikovskaya’s agar plates supplemented with 2% glucose was used to identify CaP-solubilizing bacteria from human skin. Samples from skin swabs were spread onto Pikovskaya’s agar plates, followed by a 3-day incubation. Two bacterial colonies (Figure 1 and Appendix A) developed clear zones, demonstrating their abilities to solubilize CaP. Results from 16S ribosomal RNA (rRNA) sequencing of bacterial colonies revealed that the 16S rRNA genes of these bacterial colonies shared 96.22% and 89.18% identity to that of *C. acnes* NCTC 10390 and that of *Bacillus shackletonii* (*B. shackletonii*) LMG 18435, respectively (Appendix A). The known abilities of *Bacillus* species to solubilize CaP [48,49,50,51] demonstrated the feasibility of using Pikovskaya’s agar plates to select CaP-solubilizing bacteria from human skin. The identified *C. acnes* strain [10^5^ colony-forming unit (CFU)] was incubated in phenol red-containing rich media in the presence of 2% glucose under anaerobic conditions for 10 days. Acidification of the media (4.70 vs. 7.20 and 7.15 in controls), indicated the glucose fermentation activity of this newly-identified *C. acnes* strain (Figure 1A) [38]. The glucose fermentation activity of *B. shackletonii* was confirmed as a positive control (Appendix A) [51]. The SCFAs produced by glucose fermentation of *C. acnes* were identified by gas chromatography-mass spectrometry (GC-MS) analysis [39]. Five SCFAs including acetic acid, butyric acid, propionic acid, isobutyric acid, and isovaleric acid were detectable in media of glucose fermentation of *C. acnes* (Figure 1B).

To assess whether the fermentation products of *C. acnes* can solubilize CaP, the fermented media were incubated with CaP for 24 h. The change in particle size of CaP was measured using dynamic light scattering (DLS). The particle size of CaP incubated with media from the culture of glucose or *C. acnes* alone was approximately 380 nm with an intensity of 45%. Two smaller particle sizes at 70 and 230 nm with intensities at 10 and 20% were detected when CaP was incubated with media from the glucose fermentation of *C. acnes* (Figure 1C). Acetic acid, propionic acid, or butyric acid (0–200 mM) were then added onto a Pikovskaya’s agar plates. This resulted in the formation of clear zones on the agar plate with the addition of acetic acid or propionic acid, but not butyric acid (Figure 1D). These results suggest that CaP can be degraded by acetic acid and propionic acid in the media of glucose fermentation of *C. acnes* [52,53,54].

### 3.2. Regulation of CaP-Induced IL-6 and Histone H3 Lysine 9 Acetylation (AcH3K9) by Butyric Acid and Butyric Acid N-[2-(2-Butyrylamino-ethoxy)-ethyl]-butyramide, BA-NH-NH-BA

The CaP deposits and increased level of IL-6 in skin and blood are the major manifestations observed in CKD patients undergoing dialysis [42,55,56]. We previously demonstrated that intradermal injection of CaP into mice elevated IL-6 in the skin [42]. To examine whether CaP regulates IL-6 in keratinocytes, KERTr cells were treated with 5 mg/mL CaP. The levels of IL-6 mRNA and protein were detected by real-time PCR (RT-PCR) and enzyme-linked immunosorbent assay (ELISA), respectively. As shown in Figure 2A–E, both mRNA and protein were markedly up-regulated when cells were treated with CaP for 24 h. To investigate the effect of SCFAs on CaP-induced IL-6 up-regulation, KERTr cells were treated with 4 mM acetic acid, propionic acid, or butyric acid in the presence or absence of CaP. Butyric acid, but not acetic acid or propionic acid, effectively diminished the CaP-induced IL-6 up-regulation in KERTr cells. Cumulatively, we conclude that acetic acid and propionic acid can efficiently solubilize CaP while butyric acid can attenuate CaP-induced IL-6, although it cannot solubilize CaP efficiently at physiologic temperatures and concentrations (Figure 1D and Figure 2B,C).

Butyric acid treatment has been associated with decreases in the production of cytokines such as IL-6, IL-8, tumor necrosis factor (TNF)-α, and transforming growth factor (TGF)-β, thus identifying butyric acid as a potential therapeutic agent for the reduction of inflammation [57]. Although some SCFAs have been deemed safe for consumption by the US Food and Drug Administration (FDA) [58,59], they have not yet been used for the treatment of human diseases. A short half-life is one of the major drawbacks of butyric acid as a therapeutic agent [60,61]. In this regard, we have synthesized BA-NH-NH-BA, a butyric acid derivative, by conjugation of two butyric acids to both ends of a -NH-O-NH- linker (Appendix A). We have previously demonstrated that, like butyric acid, BA-NH-NH-BA inhibits HDAC and induces AcH3K9 in keratinocytes [41]. Here we show that treatment of KERTr cells with 4 mM BA-NH-NH-BA markedly attenuated CaP-induced IL-6 at both the mRNA and protein levels (Figure 2D,E). To confirm that BA-NH-NH-BA induces AcH3K9 in keratinocytes in vivo, single-cell suspensions were prepared from biopsies of C57BL/6 mouse skin treated with acetic acid, butyric acid or BA-NH-NH-BA for 2 h. Cells were then stained with antibodies to keratin 10 (K10), a marker for differentiated keratinocytes, and antibodies to AcH3K9 for flow cytometry analysis (Figure 2F). This analysis indicated that the level of AcH3K9 in K10-positive cells from mouse skin was markedly increased when upon topical application of butyric acid or BA-NH-NH-BA, but not acetic acid (Figure 2G). This effect was confirmed by immunostaining data, which clearly showed the induction of AcH3K9 in KERTr cells treated with 4 mM butyric acid or BA-NH-NH-BA (Appendix A). All data above suggest that BA-NH-NH-BA may ameliorate CaP-induced IL-6 expression via inhibition of HDAC in keratinocytes.

### 3.3. Inhibition of CaP-Induced Itching, IL-6, and p-ERK 1/2 by BA-NH-NH-BA in Mice

Our previous study demonstrated that prolonged CaP deposition led to skin itching and an upregulated level of IL-6 in mice [42]. To investigate the effect of glucose fermentation of *C. acnes* on the action of CaP, the dorsal skin of ICR mice were intradermally injected 5 mg/mL CaP before topical application of *C. acnes* (10^5^ CFU) with and without 2 % glucose for 3 days. Injection of CaP evoked a significant increase in scratching behavior in mice topically applied with *C. acnes* alone. However, a reduction in scratching was detected in mice treated with *C. acnes* plus glucose (Figure 3A). Chronic itch sensation resulting from skin inflammation is delivered from the skin to the DRG, where ERK activation plays a vital role in spinal itch processing [42]. Accordingly, CaP-treatment resulted in the induction of IL-6 up-regulation in the skin and p-ERK 1/2 activation in the DRG in mice topically applied with *C. acnes* alone (Figure 3B–D). By contrast, an approximately 3-fold decrease in lL-6 level in the skin and p-ERK 1/2 in DRG was observed upon application of *C. acnes* along with glucose (Figure 3B–D). Furthermore, a robust increase in CaP-induced scratching (Figure 3E), the levels of IL-6 (Figure 3F), p-ERK 1/2 in DRG (Figure 3G,H) were significantly ameliorated by the topical application of 4 mM butyric acid or BA-NH-NH-BA onto CaP-injected skin. These results demonstrate the inhibitory effects of *C. acnes* fermentation, butyric acid and BA-NH-NH-BA on CaP-induced skin itching. We next used IL-6 KO mice to investigate the essential role of IL-6 in the CaP-induced increase in p-ERK 1/2. Naïve WT and IL-6 KO mice express approximately equal amounts of p-ERK 1/2, but after CaP injection, a dramatic increase in p-ERK 1/2 was observed in WT, but not IL-6 KO mice (Figure 3I,J), confirming that IL-6 mediates the CaP-induced up-regulation of p-ERK 1/2.

### 3.4. Reduction of the Abundance of Genus Cutibacterium in Itchy Skin of Patients with CKD

The relative abundance of skin bacteria at the genus level was determined by detecting 16S rRNA expression using next-generation sequencing (NGS) analysis from skin swab samples of non-itchy and itchy skin of CKD patients. Taxonomical assignment and relative abundance estimates for all detected operational taxonomic units (OTUs) for 14 bacterial genera were determined (Figure 4A). This analysis shows that the abundance of genus *Cutibacterium* in itchy skin was lower than that in the non-itchy skin of patients. A previous study demonstrated that genus *Cutibacterium* enrichment coincided with a decrease in overall microbial diversity [38,62]. In addition, the abundance of *C. acnes* in the skin is also known to decline with age [63,64]. We used RT-PCR analysis to determine the relative abundance of *C. acnes* relative to *S. epidermidis* (ATCC 12228), a commensal skin bacterium, in itchy and non-itchy skin of CKD patients. No significant difference was found in relative *C. acnes* counts in non-itchy skin of patients with CKD and healthy subjects (Figure 4B). However, the relative count of *C. acnes* was significantly lower in the itchy skin of CKD patients.

## 4. Discussion

Previous results from our laboratory have uncovered the involvement of IL-6 and p-ERK in CaP-induced pruritus in mice and a higher amount of IL-6 in itchy skin of CKD patients [42]. Here, we reveal the lower abundance of *C. acnes* in itchy skin of CKD patients and provide additional evidence validating that beneficial activity of *C. acnes* to relieve skin itch in mice. The glucose fermentation of *C. acnes* yielded large amounts of acetic acid and propionic acid, which can degrade CaP. Fetuin-A is a critical inhibitor of vascular and ectopic CaP deposition, which declined in the circulation of the dialysis population [65]. However, a low level of fetuin-A in skin throughout the epidermal and dermal layers among adults, especially with age [66,67], was detected. Our results in Figure 1 and Figure 3 showed SCFAs produced from fermentation of skin bacteria degraded CaP or reduced CaP-induced inflammation. These data suggest that Fetuin-A and SCFAs play crucial roles in preventing CaP deposition in the bloodstream and skin, respectively. It has been documented that an altered gut microbiota composition with lowered abundance for butyric acid-producing microbes and higher plasma levels of p-cresyl sulfate were detected in CKD patients. This could be overcome by dietary treatments with fibers leading to improved metabolome profile and efficient synthesis of SCFAs [68,69]. In the current study, the glucose fermentation of *C. acnes* or butyric acid in fermentation media exerts an advantageous influence on the reduction of CaP-induced scratching and up-regulation of IL-6 in the skin and p-ERK 1/2 in DRG. BA-NH-NH-BA, a butyric acid derivative, may inhibit HDAC to activate AcH3K9 in keratinocytes to block the signaling of CaP-induced skin itching mediated by IL-6 and p-ERK 1/2 (Figure 3). Since IL-6 mediated the CaP-induced skin pruritus and p-ERK 1/2 activation in DRG (Figure 3), the use of BA-NH-NH-BA, a butyric acid analog, for suppression of CaP-induced pruritic inflammation may provide a new modality for the treatment of the pruritic condition in CKD.

We have previously detected that CaP deposition in the skin induced the up-regulation of IL-6 and p-ERK 1/2 in mice [42]. To our knowledge, the current study is the first to demonstrate that skin *C. acnes* could mediate fermentation to produce butyric acid and regulate CaP-induced itch signaling by epigenetic modification. Notably, butyric acid did not influence the growth of *C. acnes* in vitro (Appendix A), suggesting that butyric acid is acting as a postbiotic rather than a microbial modulator. IL-6 is likely to be an essential mediator of this inflammatory process, since the mRNA expression of IL-6, but not IL-31 (data not shown), was up-regulated in CaP-induced skin of mice. An increase in IL-6 level in CaP-deposited skin is more likely to be associated with immune-inflammatory activation in CKD [42]. CaP-induced up-regulation of IL-6 was significantly inhibited by butyric acid, inferring that metabolites from *C. acnes* fermentation play a crucial role in the reduction of inflammation [38,70,71]. Previous studies demonstrated that pharmacological inhibition of histone deacetylases (HDAC) exerted anti-fibrosis, anti-inflammation, and immunosuppressant effects in various CKDs, showing that the inhibition of HDAC could be a logical target in pharmacotherapy against inflammation in CKD [72,73]. Butyric acid is well-established as a broad-spectrum HDAC inhibitor, which impedes most HDACs (except class III HDACs and class II HDACs 6 and 10). This is likely the principal mechanism through which it suppresses nuclear factor κB (NF-κB) activation and exerts a net anti-inflammatory effect [74]. 

As shown in Figure 2, activation of AcH3K9 is detectable in KERTr cells or mouse skin treated with butyric acid or BA-NH-NH-BA, suggesting that butyric acid or BA-NH-NH-BA may down-regulate the production of IL-6 via the inhibition of HDACs favoring histone acetylation [75]. A modeling study demonstrated that trichostatin A (TSA) exerts its inhibitory effect on HDAC inhibitor by disrupting interactions between the lysine side chain of histone H3 and the acetyl group of HDAC [76]. This may provide a model for the interactions between HDACs and the aliphatic chain and hydroxamic acid groups of butyric acid or BA-NH-NH-BA.

It has been shown that a change in the relative abundance of commensurate skin bacteria and an imbalance of Th1 and Th2 activities with upregulated inflammatory mediators such as IL-6 and IL-17 worsened the inflamed skin of AD and psoriasis. Treatment of these skin inflammatory disorders with probiotics and/or prebiotics restored the microbiome dysbiosis and relieved the inflammation [32,33,77,78,79,80,81]. Interestingly, an altered gut microbiome composition with a decrease in numbers of butyric acid-producing bacteria was also detected in CKD patients [82]. From a clinical perspective, however, it is important to note that CKD patients receiving hemodialysis may be susceptible to bacterial infection. Therefore, the topical application of live probiotic bacteria may not be the best option for the treatment of pruritus. Administration of prebiotic could be an alternative modality to boost the fermentation of *C. acnes*, but a decline of *C. acnes* in itchy skin of CKD patients creates a challenge for the proper formulation of prebiotics. The prevalence of fungal infection by onychomycosis or candidemia by Candida species could be one of the reasons for morbidity and mortality in patients undergoing hemodialysis. Previous results have demonstrated that butyric acid effectively suppressed the germination in Candida species; depicting metabolites of skin bacteria may provide a beneficial effect against microbial dysbiosis [83,84]. Our study identifies a more desirable alternative: administration of a postbiotic mimetic. Topical application of BA-NH-NH-BA onto the skin of CaP-injected mice significantly resulted in a decrease in pruritus with significant down-regulation of IL-6, highlighting the potential of developing fermentation metabolites as postbiotics for treatments of uremic pruritus [85,86].

By analyzing the microbiota composition among 14 genera of CKD patients with itchy and non-itchy conditions, we found a remarkable reduction in the abundance of genus *Cutibacterium* in itchy skin. On the other hand, a higher percentage of genus *Staphylococcus* in the majority of itchy skin of CKD patients, supporting the previous findings that members of the *genus Staphylococcus* frequently colonize itchy skin [87,88]. Results using multilocus sequence typing (MLST) and ribotyping analysis have shown that *C. acnes* with phylotype IA1/IC and III was globally associated with the severity of acne vulgaris, while phylotype II mostly colonizes healthy skin [89,90]. The constructed phylogenetic tree in Appendix A demonstrates that *C. acnes* we isolated from human skin shares the same branch and branch length with *C. acnes* phylotype II. In the analysis of the *C. acnes* abundance in CKD patients over the age of 50, itchy skin harbored a lower abundance of genus *Cutibacterium* and decreased *C. acnes* counts compared to non-itchy skin of patients and skin of healthy subjects (Figure 4). 

While CaP is an important driver of pruritus in CKD patients, other factors like a disrupted ecological niche with reduced sebum production and lower hydration of stratum corneum in the xerotic skin of uremic pruritus patients have been reported [91,92]. In addition, the concentration of urea was found to be higher in the blood circulation of CKD patients, which ultimately converts to caustic ammonium hydroxide (NH_4_OH) by urease-producing bacteria resulting in an alteration of microbiome, which subsequently decreased the production of SCFAs [93]. Furthermore, it has been documented that calcium and phosphate influenced the growth of cutaneous *C. acnes* by increasing its generation time [94].

Overall, data in our publication [42] have shown that CaP-induced pruritus was mediated by the IL-6/p-ERK signaling in mice. Itchy skin of CKD patients expressed higher amounts of IL-6. In this study, we provide the data to demonstrate that skin *C. acnes* plays a pivotal role in the signaling of CaP-induced pruritus mediated by IL-6/p-ERK. Glucose fermentation of *C. acnes* yielded CaP-solubilizing SCFAs such as propionic acid, acetic acid and anti-inflammatory butyric acid. Like butyric acid, BA-NH-NH-BA, a novel drug derived from butyric acid, may function as an HDAC inhibitor to block CaP-induced pruritus via activation of AcH3K9 in keratinocytes. Previous data from GC analysis showed that a solution with 4 mM BA–NH–NH–BA could be stored at 4 °C for six months without degradation, demonstrating its stability in vitro [41,60,61]. Future works will include evaluation of pharmacokinetics and stability of BA–NH–NH–BA in skin. Certain subtypes of opportunistic skin *C. acnes* bacteria are predominantly present in lesions of skin disorders such as acne vulgaris [89,90]. However, it has been illustrated that the abundance of *C. acnes* declined in dry and/or aged skin [63,64,95]. Our results demonstrated that itchy skin of CKD patients aged over 50 harbored a relatively low abundance of *C. acnes* (Figure 4B). Data above support the beneficial effect of *C. acnes* against pruritus.

## 5. Conclusions

Lower *C. acnes* count with increased IL-6 level are major problems in CKD patients with pruritus. The glucose fermentation of *C. acnes* or its fermentation metabolite butyric acid and its derivative BA-NH-NH-BA may inhibit HDAC to activate AcH3K9 in keratinocytes, successfully ameliorating the CaP-induced pruritic inflammation and neuropathy by inhibition of IL-6 in the skin and p-ERK 1/2 in DRG. BA-NH-NH-BA thus holds promise for improved uremic pruritus in CKD. 

## Figures and Tables

**Figure 1 jcm-09-00312-f001:**
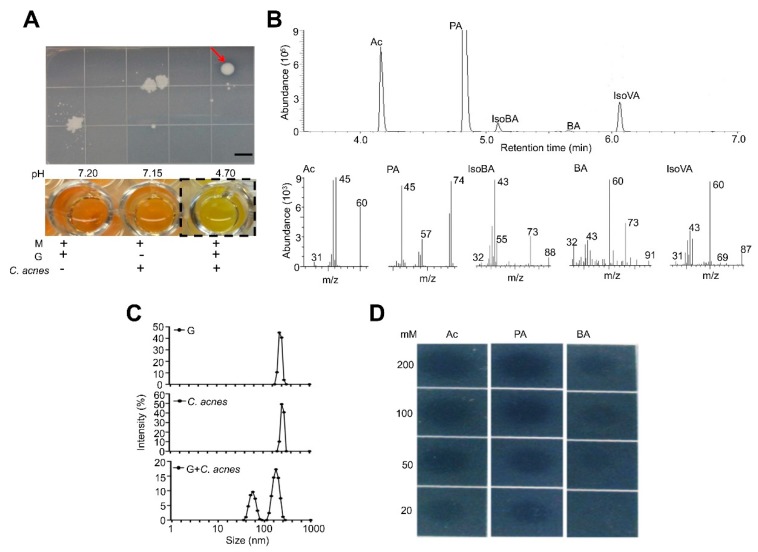
The CaP-solubilizing property of *C. acnes* and short-chain fatty acids (SCFAs) in fermentation media. (**A**) A CaP-solubilizing skin microorganism developing a clear zone (red arrow) was identified as *C. acnes*, which used 2% glucose (G) in a CaP-rich Pikovskaya’s agar to undergo fermentation. The pH values of phenol red-containing media with glucose, *C. acnes*, and glucose plus *C. acnes* were 7.20, 7.15 and 4.70, respectively. Scale bar = 1 cm. (**B**) Spectra of gas chromatography (GC) and mass spectrometry (MS) of five SCFAs produced by glucose fermentation of *C. acnes* were displayed. Fragment ions (m/z) for each SCFA were indicated. Ac, acetic acid; BA, butyric acid; IsoBA, isobutyric acid; IsoVA, isovaleric acid; PA, propionic acid. (**C**) The size of CaP was determined by dynamic light scattering (DLS) after incubation of CaP with culture media of glucose alone, *C. acnes* alone or glucose plus *C. acnes*. (**D**) Different concentrations of Ac, BA, and PA were added onto the CaP-rich Pikovskaya’s agar plates. Representative data from three independent experiments are shown.

**Figure 2 jcm-09-00312-f002:**
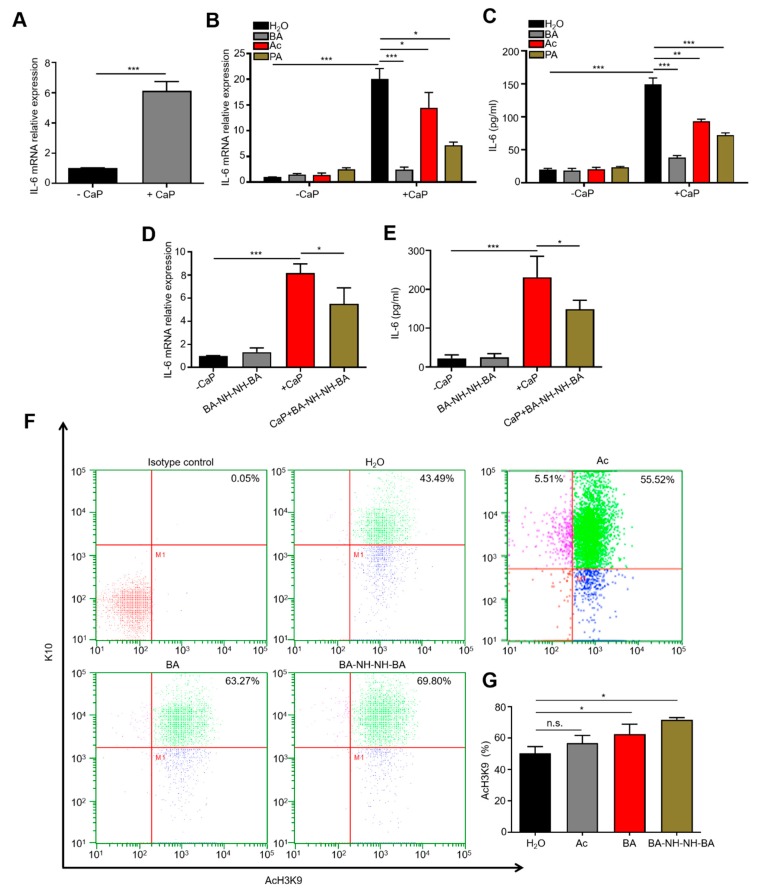
Inhibition of CaP-induced interleukin (IL)-6 in KERTr cells and induction of AcH3K9 in mouse skin by butyric acid or BA-NH-NH-BA. (**A**) mRNA expression of IL-6 in KERTr cells treated with or without CaP was detected by RT-PCR. mRNA expression (**B**,**D**) and protein level (**C**,**E**) of IL-6 KERTr cells treated with or without CaP in the presence of H_2_O, acetic acid (Ac), propionic acid (PA), butyric acid (BA), or BA-NH-NH-BA were measured by RT-PCR and ELISA, respectively. (**F**) Flow cytometry was employed to detect AcH3K9 in single-cell suspensions of mouse skin treated with H_2_O, Ac, BA, or BA-NH-NH-BA. Cells were stained with antibodies to K10, a marker for differentiated keratinocytes, and AcH3K9. Isotype controls were used to identify non-specific background staining. (**G**) The percentages (%) in cells positively stained with both AcH3K9 and K-10 were quantified. The *p*-values of <0.05 (*), <0.01 (**), and <0.001 (***) from three different experiments with mean ± SE were shown; n.s. = non-significant.

**Figure 3 jcm-09-00312-f003:**
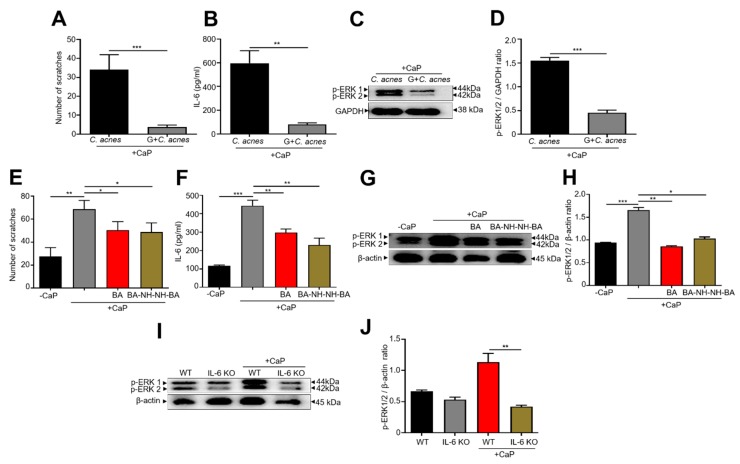
Suppression of CaP-induced itching, IL-6 production in skin and p-extracellular signal-regulated kinase (ERK) 1/2 activation in dorsal root ganglion (DRG) of mice by *C. acnes* glucose fermentation, butyric acid, or BA-NH-NH-BA. (**A**) The number of hind paw scratches was counted for 2 h after intradermal injection of CaP and topical application of *C. acnes* with and without glucose (G). (**B**) Protein level of IL-6 in CaP-injected dorsal skin of Institute for Cancer Research (ICR) mice topically applied with *C. acnes* in the presence or absence of glucose. (**C**,**D**) Western blot analysis of p-ERK 1/2 and GAPDH in the DRG of CaP-injected mice topically applied with *C. acnes* along with/without glucose. The number of hind paw scratches (**E**), IL-6 level in the skin (**F**), and p-ERK 1/2 and β-actin in the DRG (**G**,**H**) of CaP-induced mice and topically applied with and without BA or BA-NH-NH-BA were detected. Western blot analyses of p-ERK 1/2 and β-actin in the DRG (**I**,**J**) of wild type (WT) and IL-6 knockout (KO) mice injected with and without CaP. The ratios of intensities of p-ERK 1/2 relative to GAPDH (**D**) or β-actin (**H**,**J**) in western blot analysis are shown. The mean ± SE for three separate experiments was calculated. The *p*-values of < 0.05 (*), < 0.01 (**), and < 0.001 (***) were indicated.

**Figure 4 jcm-09-00312-f004:**
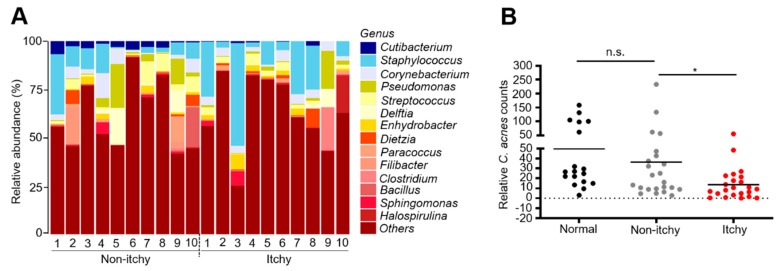
Relative abundance of skin bacteria at the genus level and *C. acnes* counts in non-itchy and itchy skin. (**A**) Next-generation sequencing (NGS) analysis of relative abundance of bacterial taxa (genus-level) in non-itchy or itchy skin of patients with chronic kidney disease (CKD). Each bar refers to a single skin swab sample and depicts the proportion of operational taxonomic units (OTUs) per sample, expressed as percentage. Color-coding of the bacterial genus is shown on the right-hand side. “Others” includes the genus of bacteria detected at a low percentage in skin swab samples. (**B**) RT-PCR analysis of *C. acnes* counts normalized to *Staphylococcus* species from non-itchy skin of healthy subjects (Normal) and the non-itchy or itchy skin of patients with CKD. The *p*-values of <0.05 (*), <0.01 (**), and <0.001 (***) from three different experiments with mean ± SE were shown; n.s. = non-significant.

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
