# Peer review of "Skin Cutibacterium acnes Mediates Fermentation to Suppress the Calcium Phosphate-Induced Itching: A Butyric Acid Derivative with Potential for Uremic Pruritus"

_jcm, 2020, doi:10.3390/jcm9020312_

Round 1
Reviewer 1 Report
The authors showed that CaP deposition activates IL-6 / p-ERK signal, causing inflammation and further itching. Furthermore, a short-chain unsaturated fatty acid produced by fermentation with C. acnes solubilize CaP and showed the possibility of relieving uremic itch.
#1 CaP (p 2, line 52)
Since the guideline to control iP, Ca and intact -PTH was published in 2006, the number of hemodialysis patients with severe ectopic calcification have decreased, and the number of hemodialysis patients with severe uremic pruritus seem have also decreased.
But In our previous study, there was no correlation between the degree of aorta calcification by CT and the degree of pruritus. (date not shown).
If there is the paper reporting that there are a different in skin CaP deposition between hemodialysis patients with itch and those without, it should be cited in the text.
#2 Bacterium (p 2, line 79-85; p 11, line 386; p11, line 397)
Previously, we cultured and examined skin bacteria from hemodialysis patients with pruritus and those without itch. There were no significant differences in the number of bacteria, Staphylococcus aureus, and Gram-negative bacteria between hemodialysis patients with pruritus and those without. However, the number of fungal colonies in the skin of hemodialysis patients with pruritus were significantly higher than that of hemodialysis patients without (date not shown).
I think that this result may be due to immune abnormalities in dialysis patients. Recently, the drug for onychomycosis in hemodialysis patients was also developed. What about your data regarding fungi on the skin of hemodialysis patients with pruritus?
#3 SCFA (p 2, line 85, p5, line 230ï¼› p 10, line 353)
It has been reported that vascular calcification and ectopic calcification in hemodialysis patients are related to the ectopic calcification inhibitor Fetuin-A.
What do you think about the relationship between Fetuin-A and acetic acid and propionic acid?
#4 IL-6, ERK1/2, etc (p2, line58; p 2, line 89; p10, line359; p12, line 415)
The chronic CaP deposition in hemodialysis patients often occurs “subcutaneously”. The authors reported that CaP-induced skin itching is mediated by IL-6 in “the skin” and p-ERK 1/2 in “DRG”, butyric acid and BA-NH-NH-BA inhibit HDAC and induces AcH3K9 in “keratinocyte (i.e., epidermis)”, and the fermentation of C. acnes occur in “the skin”.
I think the definition of skin is somewhat ambiguous.For this reason, it is difficult to understand the relationship between places where various events occur.
#5
The cause of uremic pruritus is thought to be multifactorial. Chronic CaP deposition is one of the exogenous pruritogens.
Do you think about endogenous pruritogens such as protein-binding uremic substances (e.g., p-cresyl sulfate).
Author Response
Responses to Reviewers 1
We appreciate the thoughtful and insightful comments made by the reviewers regarding our revised manuscript (jcm-679465) entitled “Skin Cutibacterium acnes mediates fermentation to suppress the calcium phosphate-induced itching: a butyric acid derivative with potential for uremic pruritus “. Changes are underlined in the text of manuscript.
Reviewer 1
The authors showed that CaP deposition activates IL-6 / p-ERK signal, causing inflammation and further itching. Furthermore, a short-chain unsaturated fatty acid produced by fermentation with C. acnes solubilize CaP and showed the possibility of relieving uremic itch.
Comment 1
CaP (p 2, line 52)
Since the guideline to control iP, Ca and intact -PTH was published in 2006, the number of hemodialysis patients with severe ectopic calcification have decreased, and the number of hemodialysis patients with severe uremic pruritus seem have also decreased. But In our previous study, there was no correlation between the degree of aorta calcification by CT and the degree of pruritus. (date not shown). If there is the paper reporting that there are a different in skin CaP deposition between hemodialysis patients with itch and those without, it should be cited in the text.
Response 1
We have added a sentence below in the second paragraph of Introduction Section and three references [2, 16, 17] has been cited.
“ Moreover, it has been reported that the levels of calcium as well as CaP in the epidermal layer of skin in hemodialysis patients with pruritus are significantly higher than those in hemodialysis patients without pruritus or healthy subjects [2,16,17].”
Berger, T.G.; Steinhoff, M. Pruritus and renal failure. Semin Cutan Med Surg 2011, 30, 99-100, doi:10.1016/j.sder.2011.04.005. Blachley, J.D.; Blankenship, D.M.; Menter, A.; Parker III, T.F.; Knochel, J.P. Uremic pruritus: skin divalent ion content and response to ultraviolet phototherapy. American Journal of Kidney Diseases 1985, 5, 237-241. Momose, A.; Kudo, S.; Sato, M.; Saito, H.; Nagai, K.; Katabira, Y.; Funyu, T. Calcium ions are abnormally distributed in the skin of haemodialysis patients with uraemic pruritus. Nephrol Dial Transplant 2004, 19, 2061-2066, doi:10.1093/ndt/gfh287.Comment 2
Bacterium (p 2, line 79-85; p 11, line 386; p11, line 397)
Previously, we cultured and examined skin bacteria from hemodialysis patients with pruritus and those without itch. There were no significant differences in the number of bacteria, Staphylococcus aureus, and Gram-negative bacteria between hemodialysis patients with pruritus and those without. However, the number of fungal colonies in the skin of hemodialysis patients with pruritus were significantly higher than that of hemodialysis patients without (date not shown). I think that this result may be due to immune abnormalities in dialysis patients. Recently, the drug for onychomycosis in hemodialysis patients was also developed. What about your data regarding fungi on the skin of hemodialysis patients with pruritus?
Response 2
We have added sentences below to the fourth paragraph of Discussion Section and two references have been cited.
“The prevalence of fungal infection by onychomycosis or candidemia by Candida species could be one of the reasons for morbidity and mortality in patients undergoing hemodialysis. Previous results have demonstrated that butyric acid effectively suppressed the germination in Candida species, depicting metabolites of skin bacteria may provide a beneficial effect against microbial dysbiosis [83,84].”
Kuvandik, G.; Cetin, M.; Genctoy, G.; Horoz, M.; Duru, M.; Akcali, C.; Satar, S.; Kiykim, A.A.; Kaya, H. The prevalance, epidemiology and risk factors for onychomycosis in hemodialysis patients. BMC Infect Dis 2007, 7, 102, doi:10.1186/1471-2334-7-102. Noverr, M.C.; Huffnagle, G.B. Regulation of Candida albicans morphogenesis by fatty acid metabolites. Infect Immun 2004, 72, 6206-6210, doi:10.1128/IAI.72.11.6206-6210.2004.Comment 3
SCFA (p 2, line 85, p5, line 230ï¼› p 10, line 353). It has been reported that vascular calcification and ectopic calcification in hemodialysis patients are related to the ectopic calcification inhibitor Fetuin-A. What do you think about the relationship between Fetuin-A and acetic acid and propionic acid?
Response 3
We have added sentences below in the first paragraph of Discussion Section and cited three references.
“Fetuin-A is a critical inhibitor of vascular and ectopic CaP deposition, which declined in the circulation of the dialysis population [65]. However, a low level of fetuin-A in skin throughout the epidermal and dermal layers among adults, especially with age [66, 67], was detected. Our results in Figures 1 and 3 showed SCFAs produced from fermentation of skin bacteria degraded CaP or reduced CaP-induced inflammation. These data suggest that Fetuin-A and SCFAs play cruial roles in preventing CaP deposition in the bloodstream and skin, respectively.
Chen, H.Y.; Chiu, Y.L.; Hsu, S.P.; Pai, M.F.; Yang, J.Y.; Peng, Y.S. Relationship between Fetuin A, Vascular Calcification and Fracture Risk in Dialysis Patients. PLoS One 2016, 11, e0158789, doi:10.1371/journal.pone.0158789. Wang, X.-Q.; Hung, B.; Kempf, M.; Liu, P.-Y.; Dalley, A.; Saunders, N.; Kimble, R. Fetuin-A promotes primary keratinocyte migration: Independent of epidermal growth factor receptor signalling. Experimental dermatology 2009, 19, e289-292, doi:10.1111/j.1600-0625.2009.00978.x. Laughlin, G.A.; Cummins, K.M.; Wassel, C.L.; Daniels, L.B.; Ix, J.H. The association of fetuin-A with cardiovascular disease mortality in older community-dwelling adults: the Rancho Bernardo study. J Am Coll Cardiol 2012, 59, 1688-1696, doi:10.1016/j.jacc.2012.01.038.Comment 4
IL-6, ERK1/2, etc (p2, line58; p 2, line 89; p10, line359; p12, line 415). The chronic CaP deposition in hemodialysis patients often occurs “subcutaneously”. The authors reported that CaP-induced skin itching is mediated by IL-6 in “the skin” and p-ERK 1/2 in “DRG”, butyric acid and BA-NH-NH-BA inhibit HDAC and induces AcH3K9 in “keratinocyte (i.e., epidermis)”, and the fermentation of C. acnes occur in “the skin”.
I think the definition of skin is somewhat ambiguous. For this reason, it is difficult to understand the relationship between places where various events occur.
Response 4
Also see the Response 1. Although calcium ion depositions were detectable in the dermal layer and had similar levels in the dermis of both chronic haemodialysis patients with and without pruritus, it has been reported that the haemodialysis patients with pruritus had significantly higher calcium ion deposition in the extracellular fluid and cytoplasm of basal cells of epidermis [17]. The result supports our finding that butryic acid produced by skin C. acnes bacteria can promote AcH3K9 and suppress the CaP-induced secretion of IL-6 from keratinocytes and activation of p-ERK 1/2 in DRG.
Momose, A.; Kudo, S.; Sato, M.; Saito, H.; Nagai, K.; Katabira, Y.; Funyu, T. Calcium ions are abnormally distributed in the skin of haemodialysis patients with uraemic pruritus. Nephrol Dial Transplant 2004, 19, 2061-2066, doi:10.1093/ndt/gfh287.
Comment 5
The cause of uremic pruritus is thought to be multifactorial. Chronic CaP deposition is one of the exogenous pruritogens.
Do you think about endogenous pruritogens such as protein-binding uremic substances (e.g., p-cresyl sulfate).
Response 5
We have added a sentence below in the first paragraph of Discussion Section and cited two references.
“It has been documented that an altered gut microbiota composition with lowered abundance for butyric acid-producing microbes and higher plasma levels of p-cresyl sulfate were detected in CKD patients. This could be overcome by dietary treatments with fibers leading to improved metabolome profile and efficient synthesis of SCFAs [68,69].
Maria De, A.; Gabriella, G.; Fabio, M.; Leonilde, B.; Piero, P.; Marco, G. The Food-gut Human Axis: The Effects of Diet on Gut Microbiota and Metabolome. Current Medicinal Chemistry 2019, 26, 3567-3583, doi:http://dx.doi.org/10.2174/0929867324666170428103848. Gryp, T.; Vanholder, R.; Vaneechoutte, M.; Glorieux, G. p-Cresyl Sulfate. Toxins (Basel) 2017, 9, doi:10.3390/toxins9020052.

Reviewer 2 Report
[Introduction]
1.A lot of sentences do not add a reference as a whole. (In normal JCM papers, one sentence ends and a reference is attached.)
- Line 38, 39, 43, 46, 50, 55, 79
2. Line 63 ~ 69: IL-6 is related to CKD, so you can refer to the Intro part, but the contents of ERK do not match the contents of this paper.
3. Line 76 ~ 78: It seems to be suitable sentence for discussion
Materials and Methods
1. ‘2.6. In Administration of CaP into cells and mice, KERTr cells were used for in vitro experiments. There are only 15 papers in pubmed. Many other papers use HaCaT cells as keratinocytes. What is the special reason for using these cells?
Results 3.1. C. acnes fermentation and CaP solubilization by fermentation metabolites]
1.Line 208 Two bacterial colonies: Is there data to support?
2. Line 209 “Results from 16S ribosomal RNA (rRNA) sequencing”: What data was used to identify the strain (Which program was used? / reference must be attached)
3. With reference to line 217, supplemental figure 2, you should attach a reference to why B.shakletonii is a control among many strains.
4. Figure 1D: It is hard to see even when printing in color
5. Lines 230-231: Since Figure 1D is difficult to see, there is insufficient evidence for this conclusion.
Results 3.2. Regulation of CaP-induced IL-6 and histone H3 lysine 9 acetylation (AcH3K9) by butyric acid and 244 butyric acid N- [2- (2-Butyrylamino-ethoxy) -ethyl] -butyramide, BA-NH-NH-BA. ]
1. In the previous sentence, related to 'supplemental figure 3' on lines 264 ~ 266, you mentioned that the short half-life of butyric acid was a problem. If so, how much is the half-life of BA-NH-NH-BA? (reference must be attached)
Results 3.3. Inhibition of CaP-induced itching, IL-6, and p-ERK 1/2 by BA-NH-NH-BA in mice.]
1. IL-6 has been confirmed to be associated with CKD. If so, is p-ERK associated with CKD or itching? (if relevant, a reference should be attached)
2. Why do you test IL-6 and p-ERK in association?
3. If you want to say that IL-6 is expressed by intracellular signal transduction, why not test all of the MAPK pathways?
[Discussion]
1.Lines 375 ~ 382: Does not seem to match the text
[References]
1. References in this paper are numbered out of order and in rule.

Author Response
Responses to Reviewers 2
We appreciate the thoughtful and insightful comments made by the reviewers regarding our revised manuscript (jcm-679465) entitled “Skin Cutibacterium acnes mediates fermentation to suppress the calcium phosphate-induced itching: a butyric acid derivative with potential for uremic pruritus “. Changes are underlined in the text of manuscript.
Reviewer 2
Introduction
Comment 1: A lot of sentences do not add a reference as a whole. (In normal JCM papers, one sentence ends and a reference is attached.)
- Line 38, 39, 43, 46, 50, 55, 79
Response 1
In Lines 38, 39, 43, 46, 50, 55 and 79, the respective references have been added at the end of each sentence.
Comment 2
Line 63 ~ 69: IL-6 is related to CKD, so you can refer to the Intro part, but the contents of ERK do not match the contents of this paper.
Response 2
For activation of p-ERK 1/2 signaling, we have cited a reference [20] regarding its activation in response to pruritogens in the spinal dorsal horn. In our recent publication [42], we have demonstrated that chronic CaP deposition in skin led to pruritus with upregulated IL-6 and p-ERK 1/2 in the DRG of the spinal dorsal horn of ICR mice. This data suggests that p-ERK 1/2 activation in spinal dorsal horn is an integral phenomenon during itch in response to pruritogens. Additionally, data from IL-6 knockout (KO) mice (Figure 3I and J) confirmed that p-ERK 1/2 activation in DRG is mediated by IL-6 in skin.
20. Zhang, L.; Jiang, G.Y.; Song, N.J.; Huang, Y.; Chen, J.Y.; Wang, Q.X.; Ding, Y.Q. Extracellular signal-regulated kinase (ERK) activation is required for itch sensation in the spinal cord. Mol Brain 2014, 7, 25, doi:10.1186/1756-6606-7-25. 42. Keshari, S.; Sipayung, A.D.; Hsieh, C.C.; Su, L.J.; Chiang, Y.R.; Chang, H.C.; Yang, W.C.; Chuang, T.H.; Chen, C.L.; Huang, C.M. The IL-6/p-BTK/p-ERK signaling mediates the calcium phosphate-induced pruritus. FASEB J 2019, 10.1096/fj.201900016RR, fj201900016RR, doi:10.1096/fj.201900016RR.Comment 3 Line 76 ~ 78: It seems to be suitable sentence for discussion
Response 3
The sentence: “Previous studies demonstrated that pharmacological inhibition of histone deacetylases (HDAC) exerted anti-fibrosis, anti-inflammation and immunosuppressant effects in various CKDs, showing that the inhibition of HDAC could be a logical target in pharmacotherapy against inflammation in CKD” has been moved from Lines 76 ~ 78 of Introduction to Discussion Section.
Materials and Methods
Comment 1
‘2.6. In Administration of CaP into cells and mice, KERTr cells were used for in vitro experiments. There are only 15 papers in pubmed. Many other papers use HaCaT cells as keratinocytes. What is the special reason for using these cells?
Response 1
Human KERTr or HaCaT cells were used because they produced plethora of cytokines including interleukin (IL)-1, -6, -7, -8, -10, -12, -15, -18, and -20, and TNF-α during skin inflammation. These cells have been widely used for studies of drug responses and skin innate immnity [43, 44]. We have cited two references.
Gröne, A. Keratinocytes and cytokines. Veterinary Immunology and Immunopathology 2002, 88, 1-12, doi:https://doi.org/10.1016/S0165-2427(02)00136-8. Olaru, F.; Jensen, L.E. Chemokine expression by human keratinocyte cell lines after activation of Toll-like receptors. Exp Dermatol 2010, 19, e314-316, doi:10.1111/j.1600-0625.2009.01026.x.Results
3.1. C. acnes fermentation and CaP solubilization by fermentation metabolites.
Comment 1
Line 208 Two bacterial colonies: Is there data to support?
Response 1
In the Line 208, the colonies of two bacteria (C. acnes and Bacillus shacketonii) in Figure 1 and Supplemental Figure 1 were identified from a Pikovskaya’s agar plate. We have corrected the sentence in Line 208 as below:
Two bacterial colonies (Figure 1 and Supplemental Figure 1) developed clear zones, demonstrating their abilities to solubilize CaP.
References for the known abilities of Bacillus species to solubilize CaP have been cited [48-51].
Comment 2
Line 209 “Results from 16S ribosomal RNA (rRNA) sequencing”: What data was used to identify the strain (Which program was used? / reference must be attached)
Response 2
We have added the details for data analyized by 16S ribosomal RNA (rRNA) sequencing in the Supplementary file as described below. One reference [1] has been cited.
The 16S rRNA gene of C. acnes isolated from human skin was sequenced using an Applied Biosystems 3730xl DNA Analyzer, USA [1]. The sequences were aligned with 16S rRNA sequences from various subtypes of C. acnes including ATCC 6919 (GenBank accession no. AB042288.1), KPA171202 (GenBank accession no. AE017283.1), NCTC 10390 (GenBank accession no. AY642044.1) and Asn12 (GenBank accession no. DQ672259.1). The phylogenetic relationships were determined by using the Data Analysis in Molecular Biology and Evolution (DAMBE) software (https://www.ebi.ac.uk/Tools/msa/muscle/). Multiple sequence alignments (MSA) were performed by using the CLUSTAL W algorithm [2,3] and exported into the DAMBE program. The phylogenetic tree was constructed by the maximum-parsimony method.
Grice, E.A.; Kong, H.H.; Renaud, G.; Young, A.C.; Program, N.C.S.; Bouffard, G.G.; Blakesley, R.W.; Wolfsberg, T.G.; Turner, M.L.; Segre, J.A. A diversity profile of the human skin microbiota. Genome Res 2008, 18, 1043-1050, doi:10.1101/gr.075549.107Comment 3
With reference to line 217, supplemental figure 2, you should attach a reference to why B. shakletonii is a control among many strains.
Response 3
We have attached a reference to Line 217, Supplemental Figure 2, to support that B. shackletonii was used as a control for its ability to solubilize tricalcium phosphate.
Jikare, A.; Chavan, M. Siderophore produced by Bacillus shackletonii. Gn-09 and showed its plant growth promoting activity. Int. J. Pharm. Biol. Sci 2013, 3, 198-202.Comment 4
Figure 1D: It is hard to see even when printing in color
Response 4
A clear photo in Figure 1D has been included.
Comment 5
Lines 230-231: Since Figure 1D is difficult to see, there is insufficient evidence for this conclusion.
Response 5
A clear photo in Figure 1D has been included.
We have cited two new references to support this data.
Sharon, J.; Hathwaik, L.; Glenn, G.M.; Imam, S.; Lee, C.C. Isolation of efficient phosphate solubilizing bacteria capable of enhancing tomato plant growth. Journal of soil science and plant nutrition 2016, 16, doi:10.4067/S0718-95162016005000043. Chen, Y.; Rekha, P.; Arun, A.; Shen, F.; Lai, W.-A.; Young, C. Phosphate solubilizing bacteria from subtropical soil and their tricalcium phosphate solubilizing abilities. Applied soil ecology 2006, 34, 33-41.Results
3.2. Regulation of CaP-induced IL-6 and histone H3 lysine 9 acetylation (AcH3K9) by butyric acid and 244 butyric acid N- [2- (2-Butyrylamino-ethoxy) -ethyl] -butyramide, BA-NH-NH-BA.]
Comment 1
In the previous sentence, related to 'supplemental figure 3' on lines 264 ~ 266, you mentioned that the short half-life of butyric acid was a problem. If so, how much is the half-life of BA-NH-NH-BA? (reference must be attached)
Response 1
Regarding the stability of BA-NH-NH-BA, we have added the sentence below in the end paragraph of Discussion Section and cited three references.
Previous data from GC analysis showed that a solution with 4 mM BA–NH–NH–BA can be stored at 4°C for six months without degradation, demonstrating its stability in vitro [41,60,61]. Future works will include evaluation of pharmacokinetics and stability of BA–NH–NH–BA in skin.
41. Traisaeng, S.; Herr, D.R.; Kao, H.J.; Chuang, T.H.; Huang, C.M. A Derivative of Butyric Acid, the Fermentation Metabolite of Staphylococcus epidermidis, Inhibits the Growth of a Staphylococcus aureus Strain Isolated from Atopic Dermatitis Patients. Toxins (Basel) 2019, 11, doi:10.3390/toxins11060311. 60. Banasiewicz, T.; Krokowicz, L.; Stojcev, Z.; Kaczmarek, B.F.; Kaczmarek, E.; Maik, J.; Marciniak, R.; Krokowicz, P.; Walkowiak, J.; Drews, M. Microencapsulated sodium butyrate reduces the frequency of abdominal pain in patients with irritable bowel syndrome. Colorectal Dis 2013, 15, 204-209, doi:10.1111/j.1463-1318.2012.03152.x. 61. Pituch, A.; Walkowiak, J.; Banaszkiewicz, A. Butyric acid in functional constipation. Gastroenterology Review/PrzeglÄ…d Gastroenterologiczny 2013, 8, 295-298, doi:10.5114/pg.2013.38731.Results
3.3. Inhibition of CaP-induced itching, IL-6, and p-ERK 1/2 by BA-NH-NH-BA in mice.
Comment 1
IL-6 has been confirmed to be associated with CKD. If so, is p-ERK associated with CKD or itching? (if relevant, a reference should be attached)
Response 1
See Response above.
For activation of p-ERK 1/2 signaling, we have cited a reference [20] regarding its activation in response to pruritogens in the spinal dorsal horn. In our recent publication [42], we have demonstrated that chronic CaP deposition in skin led to pruritus with upregulated IL-6 and p-ERK 1/2 in the DRG of the spinal dorsal horn of ICR mice. This data suggests that p-ERK 1/2 activation in spinal dorsal horn is an integral phenomenon during itch in response to pruritogens. Additionally, data from IL-6 knockout (KO) mice (Figure 3I and J) confirmed that p-ERK 1/2 activation in DRG is mediated by IL-6 in skin.
Two references [20, 42] have been cited.
Comment 2
Why do you test IL-6 and p-ERK in association?
Response 2
Also see Responses above with references [20, 42]. Literature has shown that activation of p-ERK signaling is an integral phenomenon in elevating spinal neuron excitability in response to pruritogens. Results in our publication [42] showed activation of IL-6 in skin and p-ERK in DRG can be induced by on CaP in a dose manner in skin of ICR mice. Additionally, results using IL-6 knockout (KO) mice (Figure 3I and J) confirmed that p-ERK 1/2 activiation in DRG is mediated by IL-6 in skin. leading to chronic itch sensation in CKD.
Comment 3
If you want to say that IL-6 is expressed by intracellular signal transduction, why not test all of the MAPK pathways?
Response 3
We here have detected the activiation of p-ERK signaling in CaP-induced IL-6 pruritus. Studies on other MAPK pathways will be conducted in the future.
[Discussion]
Comment 1 Lines 375 ~ 382: Does not seem to match the text
Response 1
To match description in the Lines 375 ~ 382 (from Result Section) with Discussion Section, we have added one sentence in first paragraph of Discussion Section as described below:
“Since IL-6 mediated the CaP-induced skin pruritus and p-ERK 1/2 activation in DRG (Figure 3), the use of BA-NH-NH-BA, a butyric acid analog, for suppression of CaP-induced pruritic inflammation may provide a new modality for treatment of the pruritic condition in CKD.”
[References]
Comment 1: References in this paper are numbered out of order and in rule.
Response 1
The order of reference number has been fixed.
